# On Optimal Early Stopping: Overparametrization versus Underparametrization

## Abstract

Early stopping is a simple and widely used method to prevent over-training neural networks. We develop theoretical results to reveal the relationship between optimal early stopping time and model dimension as well as sample size of the dataset for certain linear regression models. Our results demonstrate two very different behaviors when the model dimension exceeds the number of features versus the opposite scenario. While most previous works on linear models focus on the latter setting, we observe that in common deep learning tasks, the dimension of the model often exceeds the number of features arising from data. We demonstrate experimentally that our theoretical results on optimal early stopping time corresponds to the training process of deep neural network. Moreover, we study the effect of early stopping on generalization and demonstrate that optimal early stopping can help mitigate "double descent" in various settings.

## 1 Introduction

Generalization, accuracy, and computation are three of the major aspects in deploying large scale machine learning models. They are often brought together into an optimization framework, where the first two aspects concern stationary solution and the last aspect concerns convergence properties of the algorithm. As a result, most previous works have analyzed them separately, seeking to strike a balance between generalization and accuracy first then understand computational complexity (Zhang, 2002). These approaches often design a regularizer in the form of a penalty term added to the objective function (c.f., Nakkiran et al., 2020b, and references therein). While this method is effective in many settings, it can prove challenging to determine the regularization that guarantees appealing behaviors. Adding to the complication is the fact that they are often designed to avoid overfitting when the optimization algorithm converges. It is not obvious how the behavior will change if the algorithm stops before the convergence of the optimization procedure.

Recent success of deep learning models combined with gradient based optimization has prompted increasingly many works to focus on blending the computation aspect into the picture of generalization–accuracy tradeoff (Du et al., 2018). Such approaches often leverage implicit properties of the optimization algorithms in conjunction of the model and the data structures and are thus referred to as implicit regularization methods. In training of deep neural networks, techniques such as stochastic gradient descent, batch normalization (Ioffe & Szegedy, 2015), and dropouts (Srivastava et al., 2014) are widely used to grant generalization properties implicitly.

In this paper, we study regularization via early stopping (Morgan & Bourlard, 1989; Zhang & Yu, 2005; Yao et al., 2007). While in practice, early stopping is often forced by computational budget constraints, we focus on its efficacy in avoiding overfitting towards the training dataset. In the presence of label noise, in particular, we study the common strategy of concluding the training process at the optimal early stopping time, which achieves the lowest test risk before it increases again. One crucial question is how this optimal early stopping time relates to the model size and sample size. Answering this question not only provides guidance for the model training process in practice, but also contributes to the understanding of the generalization property of different models.

**A Conundrum on Optimal Early Stopping**    (Ali et al., 2019) studied the regularization effect of early stopping on least square regression, but did not provide explicit characterization of the optimal early stopping time with respect to the model and data complexity. As a first step, we

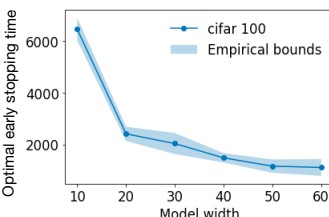 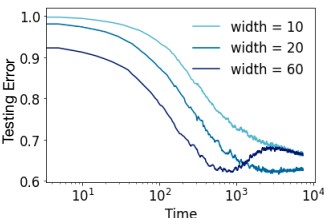 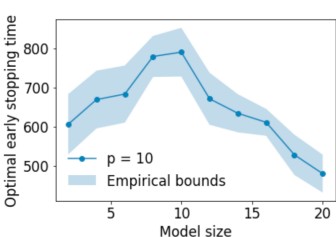

Figure 1: Left: Optimal early stopping time vs. model width $d$ for CIFAR-100 with 20% label noise trained using ResNet networks with convolutional layers of widths $[d, 2d, 4d, 8d]$. Right: Testing error vs. time for CIFAR-100 with 20% label noise trained using ResNet networks using different model widths.

Figure 2: Optimal early stopping time vs. hidden layer width trained using two-layer ReLU networks on randomly generated points of $p = 10$ clusters with 20% label noise.

compute the optimal early stopping time of the model studied in (Ali et al., 2019) and discover that under the setting adopted therein, the optimal early stopping time increases along with the model size. However, this trend is contrary to what we observed in traning neural networks. We observed that when training popular datasets such as MNIST, CIFAR-10/100 using deep neural networks, the optimal early stopping time decreases as the model size increases (see Figure 1). These two contradicting trends indicate that previous theoretical models are not sufficient to explain what happens in practice.

**Overparametrization vs. Underparametrization** We noticed that despite the huge models used to train those datasets, their labels are actually generated by low dimensional representations or features (See Section 4.2). Previous theoretical works have primarily focused on the case where the model dimension is approximately the same or less than the number of features instead of the opposite scenario (Ali et al., 2019; Belkin et al., 2020; Nakkiran et al., 2020b). To complete the missing pieces in previous analyses, we propose a new model where the label is generated by a low-dimensional transformation of the input data. We define the case where the model size is larger than the number of features as being underparametrized. We verify experimentally that this overparametrized case better reflects what happens in training neural networks. This new overparametrization setting demonstrates very different behavior from the previous theoretical model, which resolves the contradiction we previously observed.

We illustrate both theoretically and experimentally that the optimal stopping time behaves very differently when the number of features is smaller (overparametrization) or larger (underparametrization) than the model size. In the overparametrization setting, the optimal early stopping time decreases as the model dimension increases or as the sample size decreases, similar to what we observe when training deep neural networks for image classification tasks. On the other hand, in the underparametrization setting, the optimal early stopping time increases as the model dimension increases or as the number of samples increases.We corroborate experimentally that these trends persist when training various linear models and neural networks (e.g., Figure 2).

**Generalization of the early stopped-models** Even for the same model, stopping at the optimal early stopping time versus after the training converges can lead to solutions with very different generalization properties. Another problem we study is how the test risk of models changes as a function of other training parameters when stopped at the optimal early stopping time. In particular, it has been shown that without regularization, the test risk can exhibit the so-called "double descent" phenomenon (Belkin et al., 2018; Nakkiran et al., 2020a), where the risk as a function of model size or sample size experience two distinct phases of descent. This phenomenon can make deciding the best model parameters challenging. We demonstrate that early stopping helps mitigate double descent in multiple settings.

## 1.1 RELATED WORKS

The test risk of simplified machine learning models has been studied in a long line of works. A partial list of papers that studied linear models similar to ours includes (Bartlett et al., 2020; Chen et al.,

2020a; Dobriban et al., 2018; Hastie et al., 2019; Montanari et al., 2019; Mitra, 2019; Muthukumar et al., 2020)

Regularization is widely used in training machine learning models to prevent overfitting. (Dobriban et al., 2018; Dobriban & Sheng, 2020; Hastie et al., 2019; Kobak et al., 2020; Mei & Montanari, 2019; Nakkiran et al., 2020b) studied the test risk of ridge regression. In addition, a lot of recent works studied the theoretical guarantee of implicit regularization in model training (Dereziński et al., 2019; Vaskevicius et al., 2019; HaoChen et al., 2020; Blanc et al., 2020; Razin & Cohen, 2020). In our work, we focus on regularization using early stopping. (Li et al., 2020) proved that early stopping is robust to label noise in certain settings. (Ali et al., 2019; 2020) related the risk of ridge regression to that of early stopping. (Vaškevičius et al., 2020) studied early stopped mirror descent. Our work characterizes the time that achieves the optimal stopping in various settings and demonstrates how optimal early stopping affects generalization.

The double descent phenomenon has a long history of being studied in the literature. (Loog et al., 2020) gave a brief history of the early works studying this topic. A partial list of works that tackled related problem in the setting of least square regression includes (Belkin et al., 2018; 2020; Bartlett et al., 2020; Deng et al., 2019; Hastie et al., 2019; Mei & Montanari, 2019; Muthukumar et al., 2020; d'Ascoli et al., 2020; Chen et al., 2020a; Yang et al., 2020). (Nakkiran et al., 2020a) demonstrated double descent can also occur as a function of number of samples. (Nakkiran et al., 2020b) shows that optimal regularization can mitigate double descent. Our work differs from previous works by studying the effect of early stopping on double descent.

Finally, our work studies the low rank structure of the feature representations. (Li et al., 2018; Dusenberry et al., 2020) have also empirically explored and leveraged this property.

## 2 OVERPARAMETRIZATION

In this section, we introduce and analyze the overparametrization setting, where the model size exceeds the number of features. This setting completes the missing piece of previous theoretical analysis on large models with excessive parameters. For data $(x, y)$, the input $x \in \mathbb{R}^d$ is from a high dimensional space. The label $y \in \mathbb{R}$ is generated from a low-dimensional feature representation $z \in \mathbb{R}^p$, $p \leq d$, which is mapped from $x$ by a low-dimensional transformation. A lot of common deep learning applications falls under this setting. Despite the huge model size used, the underlying feature space dimension can be low. For example, we observe that the labels $y$ in CIFAR10, a widely used image classification problem with 10 classes, can be generated by a feature space of dimension a little more than 10 (See Section 4.2).

Formally, we consider the following linear regression problem with $d > p$. Let the covariate/input matrix $X \in \mathbb{R}^{n \times d}$ be a random matrix such that each entry of it is generated from $\mathcal{N}(0, 1)$ independently. Let $x_1, ..., x_n$ be the row vectors of the matrix $X$. For a semi-orthogonal matrix $P \in \mathbb{R}^{p \times d}$ with $p < d$ and some unknown parameter $\theta^*$, the response is generated by

$$y_i = \langle Px_i, \theta^* \rangle + \epsilon_i, \tag{1}$$

where the label noise $\epsilon_i \sim \mathcal{N}(0, \sigma^2)$ is independent of $x_i$ and is generated independently for each $(x_i, y_i)$. Let $\epsilon \in \mathbb{R}^n$ be the vector with entries given by $\epsilon_i$.

We let $\mathcal{S}_{\text{over}}$ denote the overparametrization setting studied in this section. This setting is inspired by the overparametrization property of many real datasets and models. For example, to classify images using deep neural network, the labels can usually be determined by features extracted from certain important parts of the images. The other parts of the image, though do not contribute to the label much, are still input to the model. When the neural network is wide, the network has the ability to extract far more features than what contribute to the label. This character of the deep neural network is what the setting $\mathcal{S}_{\text{over}}$ attempts to model.

Our data assumption has been studied in many related literature (Belkin et al., 2020; Nakkiran et al., 2020b). Our analysis only relies on the concentration property of singular values of the data matrix $X$, so it is possible relax the data assumptions on $X$ to bounds on the minimum and maximum singular values of the data matrix (Vaswani & Nayer, 2016). The linear model has been widely used as the first step to study more complicated models as well due to its simple form (Bartlett et al., 2020; Chen et al., 2020a; Dobriban et al., 2018; Hastie et al., 2019; Montanari et al., 2019; Mitra,

2019; Muthukumar et al., 2020). It is possible to extend our results to non-linear settings such as the random feature setting. In this paper, we stick to the simplest model so that we can get a simple and explicit characterization of the optimal early stopping time.

For fixed semi-orthogonal matrix $P$ and parameter $\theta^*$, let $\mathcal{D}_{P,\theta^*}$ be the joint distribution of a single data point $(x, y)$ given $P$ and $\theta^*$. For an estimator $\beta$, define the population risk

$$R(\beta) = \mathbb{E}_{(\tilde{x},\tilde{y})\sim\mathcal{D}_{P,\theta^*}} \left[ (\langle \tilde{x}, \beta \rangle - \tilde{y})^2 \right].$$

We consider the standard linear least squares problem

$$\min_{\beta \in \mathbb{R}^d} \frac{1}{2n} \|y - X\beta\|_2^2. \tag{2}$$

Applying gradient flow starting from 0 on equation 2 gives a continuous differential equation over time $t$,

$$\frac{d}{dt}\beta_{X,y}(t) = \frac{X^\top}{n}(y - X\beta_{X,y}(t)). \tag{3}$$

with an initial condition $\beta_{X,y}(0) = 0$. We can solve the differential equation exactly and obtain

$$\beta(t) = \left(X^\top X\right)^+ \left(I - \exp\left(-tX^\top X/n\right)\right) X^\top y, \tag{4}$$

for all $t \geq 0$ (Ali et al., 2019). Here, for any matrix $A$, $\exp(A)$ is defined as $\exp(A) := \sum_{n=0}^\infty \frac{A^n}{n!}$. In practice, one would discretize the gradient flow and perform gradient descent. We discuss the error due to such discretization in Appendix E.

For any fixed dataset $y \in \mathbb{R}^n$ and $X \in \mathbb{R}^{n \times d}$ given by fixed $P$ and $\epsilon$, let $\beta_{X,y}(t)$ be the gradient flow solution of standard linear regression problem at time $t$. We further assume that $P \in \mathbb{R}^{p \times d}$ is a uniformaly random semi-orthogonal matrix.

Our goal is to study the expected risk of the estimator $\beta_{X,y}(t)$ over $P$ and $\epsilon$. We denote the expected risk at time $t$ for dataset size $n$ as

$$\overline{R}_{X,\theta^*}(t) = \mathbb{E}_{P,\epsilon}\left[R(\beta_{X,y}(t))\right].$$

We study the optimal early stopping time, the time $t$ that achieves the first local minimum of the expected risk. We characterize the optimal early stopping time for different parameter choices. We let $t_{\text{opt}}$ denote the optimal stopping time, omitting $(X, \theta^*)$ when clear from context,

$$t_{\text{opt}}(X, \theta^*) = \min t \ \ s.t. \ \ \frac{d}{dt}\overline{R}_{X,\theta^*}(t') < 0 \text{ for } 0 < t' < t, \text{ and } \frac{d}{dt}\overline{R}_{X,\theta^*}(t) = 0. \tag{5}$$

### 2.1 Optimal Early Stopping Time

With the gradient flow solution $\beta_{X,y}$, we can derive a high probability upper and lower bound on $t_{\text{opt}}$.

**Theorem 1** (Optimal early stopping time in overparametrization setting). *In setting $\mathcal{S}_{over}$, for a fixed parameter $\theta^*$ and noise variance $\sigma^2$, when $n \leq d$, let $\gamma = \left(\frac{\sqrt{n}+\sqrt{2\log n}}{\sqrt{d}}\right)^2$. For $\gamma \leq 1$, with probability at least $1 - \frac{2}{n}$ over the randomness of $X$, the optimal early stopping time $t_{\text{opt}}$ satisfies*

$$\frac{n}{\left(1+\sqrt{\gamma}\right)^2 d} \log\left(1 + \frac{\left(1-\sqrt{\gamma}\right)^2 \|\theta^*\|_2^2}{\sigma^2}\right) \leq t_{\text{opt}} \leq \frac{n}{\left(1-\sqrt{\gamma}\right)^2 d} \log\left(1 + \frac{\left(1+\sqrt{\gamma}\right)^2 \|\theta^*\|_2^2}{\sigma^2}\right).$$

*When $n > d$, let $\gamma = \left(\frac{\sqrt{n}}{\sqrt{d}+\sqrt{2\log d}}\right)^2$. For $\gamma \geq 1$, with probability at least $1-\frac{2}{d}$ over the randomness of $X$, the optimal early stopping time $t_{\text{opt}}$ satisfies*

$$\frac{1}{\left(1+\frac{1}{\sqrt{\gamma}}\right)^2} \log\left(1 + \left(1-\frac{1}{\sqrt{\gamma}}\right)^2 \frac{n\|\theta^*\|_2^2}{d\sigma^2}\right) \leq t_{\text{opt}} \leq \frac{1}{\left(1-\frac{1}{\sqrt{\gamma}}\right)^2} \log\left(1 + \left(1+\frac{1}{\sqrt{\gamma}}\right)^2 \frac{n\|\theta^*\|_2^2}{d\sigma^2}\right).$$

Note that our definitions of $\gamma$ differ slightly in the case $n \leq d$ and $n > d$, but they are both of order $\tilde{\Theta}\left(\frac{n}{d}\right)$. Theorem 1 gives an approximation of the optimal stopping time up to a constant as long as $\gamma$ is constantly bounded away from 1. When $\gamma \leq \frac{1}{4}$ (roughly $n \leq 4d$),omitting the logarithmic terms, $\frac{9n}{4d} \leq t_{\text{opt}} \leq \frac{2n}{d}$. When $\gamma \geq 4$ (roughly $n \geq 4d$), ~~omitting the logarithmic terms,~~ $\frac{n}{3(d+n)} \log\left(1 + \frac{n\|\theta^*\|}{4d\sigma^2}\right) \leq t_{\text{opt}} \leq \frac{5n}{d+n} \log\left(1 + 3\frac{n\|\theta^*\|}{d\sigma^2}\right)$. Combining these two cases shows that when $\gamma$ is bounded away from 1, $t_{\text{opt}}$ is about $\Theta\left(\frac{n}{d+n} \log \frac{n}{d}\right)$. When $\gamma$ is very close to 1, the upper bound on $t_{\text{opt}}$ in Theorem 1 can be loose. We show in Appendix C, that in the asymptotic case where $n$ and $d$ are large and $\frac{\|\theta^*\|}{\sigma^2}$ is lower bounded, the approximation $t_{\text{opt}} = \tilde{\Theta}\left(\frac{n}{d+n}\right)$ still holds when $\frac{1}{4} < \gamma < 4$. In the asymptotic analysis we also achieve tighter constants in the bounds of the optimal early stopping time. Furthermore, in Appendix F, using numerical experiments, we show that our theoretical result can be a good approximation of the optimal stopping time for all $\gamma$.

Now, we discuss some implications of Theorem 1. The following lemma computes $\overline{R}_{X,\theta^*}(t_{\text{opt}})$, the expected risk at time $t_{\text{opt}}$, and holds for all input matrix $X$ regardless of how it is generated.

**Lemma 1.** *For all $n, p, d \in \mathbb{N}$ such that $p \leq d$, let $X \in \mathbb{R}^{n \times d}$ and $\theta^* \in \mathbb{R}^p$ be fixed. Let $\lambda_1 \geq ... \geq \lambda_d$ be the eigenvalues of the matrix $\frac{1}{n}X^\top X$. Then,*

$$\overline{R}_{X,\theta^*}(t) = \sigma^2 + \sum_{i=1}^{d} \exp\left(-2t\lambda_i\right) \frac{\|\theta^*\|_2^2}{d} + \frac{\sigma^2}{n} \sum_{i=1}^{d} \mathbb{1}\{\lambda_i \neq 0\} \frac{1}{\lambda_i}\left(1 - \exp(-t\lambda_i)\right)^2. \quad (6)$$

In equation 6, the second term is the approximation error due to early stopping. The third term is due to overfitting to the label noise, which increases as the training time increases. In presence of label noise, the optimal early stopping finds the balance point of these two terms.

Theorem 1 shows that the optimal early stopping time is roughly $\tilde{\Theta}\left(\frac{n}{n+d}\right)$, when $n$ and $d$ are around the same order. When $n \gg d$, the optimal early stopping time is roughly $\Theta(\log \frac{n}{d})$. It implies that **when the number of features is smaller than the model dimension, optimal early stopping time increases as $n$ increases or $d$ decreases.** Section F.1 corroborates this trend using experiments. In addition, we show in Section 4.2 and Section 4.3 that training deep neural networks on large real datasets can also follow these two trends. An intuitive explanation of such phenomenon is when the sample size is large, the label noise has less effect on model training, so the model can be trained longer before overfitting starts. On the contrary, when $d$ increases, the number of parameters is large so that even small updates of each parameter can result in overfitting.

## 2.2 RISK MONOTONICITY

It has been observed empirically that without early stopping or other types of regularization, when the number of samples $n$ increases, the test error can experience double descent in presence of label noise. However, optimal early stopping can possibly mitigate the double descent phenomenon (Nakkiran et al., 2020a). In Proposition 1, we bound the expected risk at the optimal stopping time for varying data size $n$ and attempt to explain this phenomenon.

**Proposition 1.** *In setting $\mathcal{S}_{over}$, for a fixed parameter $\theta^*$ and noise variance $\sigma^2$. Let $\bar{t}_{opt} = \alpha n/(n+d)$ for some constant $\alpha$. Let $\lambda_1 \geq ... \geq \lambda_d$ be the eigenvalues of the matrix $\frac{1}{n}X^\top X$. Then,*

$$\overline{R}_1 \leq \mathbb{E}_X[\overline{R}_{X,\theta^*}(\bar{t}_{opt})] \leq \overline{R}_2. \quad (7)$$

*where*

$$\overline{R}_1 = \sigma^2 + \mathbb{E}_X\left[\sum_{i=1}^{d} \exp\left(-2\alpha n\lambda_i/d\right)\right] \frac{\|\theta^*\|^2}{d}, \quad and \quad \overline{R}_2 = \overline{R}_1 + \frac{1}{2}\alpha\sigma^2.$$

$\overline{R}_1$ *and* $\overline{R}_2$ *decrease monotonically as $n$ increases.*

Theorem 1 shows that the optimal early stopping time is roughly $t_{opt} \approx \frac{\alpha n}{n+d}$ for a small constant $\alpha$ up the logarithmic factors. When $\frac{\|\theta\|^*}{\sigma^2}$ is not too small, equation 7 gives a small region that bounds the expected risk at $\bar{t}_{opt}$. We are able to show that both the upper and the lower bounds of the

region decrease as $n$ increases. This implies that **optimal early stopping can mitigate sample-wise double descent.** We use experiments to corroborate our computation and give more details on this observation in Section 4 and Appendix F.

A related question is whether stopping at optimal early stopping time can mitigate the double descent due to increasing model size $d$. (Nakkiran et al., 2020a) observed that double descent can still exist at optimal early stopping. An theoretical analysis of this question can be an interesting future direction.

## 3 UNDERPARAMETRIZATION

In this section, we study the underparametrization setting, where the number of features exceeds the model size. This setting shows very different behavior from the overparametrization setting. In this setting, the label $y$ is generated by some covariate $z$, which is in a high-dimensional space, but the model only has access to a low-dimension projection of $z$.

Formally, we consider the following linear regression problem with $d \leq p$. Let the covariate matrix $Z \in \mathbb{R}^{n \times p}$ be a random matrix such that each entry of it is generated from $\mathcal{N}(0, 1)$ independently. Let $z_1, ..., z_n$ be the row vectors of the matrix $Z$. For some unknown paratmeter $\theta^*$, the reponse is generated by

$$y_i = \langle z_i, \theta^* \rangle + \epsilon_i. \tag{8}$$

where the label noise $\epsilon_i \sim \mathcal{N}(0, \sigma^2)$ is independent of $z_i$ and $y_i$. For a semi-orhotgonal matrix $P \in \mathbb{R}^{p \times d}$ with $d \leq p$, let $X = ZP$. we apply gradient flow on $(X, y)$. For fixed semi-orthogonal matrix $P$ and parameter $\theta^*$, let $\mathcal{D}_{P,\theta^*}$ be the joint distribution of a single data point $(x, y)$. For an estimator $\beta$, define the population risk

$$R(\beta) = \mathbb{E}_{(\tilde{x},\tilde{y}) \sim \mathcal{D}_{P,\theta^*}} \left[ (\langle \tilde{x}, \beta \rangle - \tilde{y})^2 \right].$$

Similar as in the overparametrizations setting, we consider the gradient flow solution (equation 4) of the standard linear least square problem (equation 2). For any fixed dataset $y \in \mathbb{R}^n$ and $X \in \mathbb{R}^{n \times d}$ given by fixed $Z$, $P$ and $\epsilon$, let $\beta_{X,y}(t)$ be the solution of the gradient flow given in equation 4. We further assume that $P \in \mathbb{R}^{p \times d}$ is a uniformly random semi-orthogonal matrix.

We let $\mathcal{S}_{under}$ denote the above setting. $\mathcal{S}_{under}$ studies the case where we only have partial access to the features that determine the label $y$ and train the model on what is available.

Our goal is to study the expected risk of the estimator $\beta_{X,y}(t)$ over $Z$, $P$ and $\epsilon$. We denote the expected risk over $Z$, $P$ and $\epsilon$ at time $t$ as

$$\overline{R}_{\theta^*}(t) = \mathbb{E}_{Z,P,\epsilon} \left[ R(\beta_{X,y}(t)) \right].$$

Note that here in $\mathcal{S}_{under}$, we take expectation on $Z$ when computing the expected risk. In $\mathcal{S}_{over}$, we do not take expectation on $X$ and instead get a high probability bound on $X$. This slight difference is due to the different methods we use in analyzing these two settings. We study the optimal early stopping time, the time $t$ that achieves the first local minimum of the expected risk. We charaterize the optimal early stopping time for different choices of $p$, $n$ and $d$. We let $t_{opt}$ denote the optimal stopping time, omitting $\theta^*$ when clear from context,

$$t_{opt}(\theta^*) = \min t \quad s.t. \quad \frac{d}{dt} \overline{R}_{\theta^*}(t') < 0 \text{ for } 0 < t' < t, \text{ and } \frac{d}{dt} \overline{R}_{\theta^*}(t) = 0. \tag{9}$$

We are able to give a linear approximation of the optimal stopping time, $t_{opt}$.

**Theorem 2** (Optimal early stopping time in underparametrization setting). *In setting $\mathcal{S}_{under}$, for a fixed parameter $\theta^*$ and noise variance $\sigma^2$, if $(8n + 9d + 16) \|\theta^*\|_2^2 \leq p\sigma^2 + p \|\theta^*\|_2^2$, the optimal early stopping time $t_{opt}$ satisfies*

$$\frac{n \|\theta^*\|_2^2}{2 \left( p\sigma^2 + (p - d) \|\theta^*\|_2^2 \right)} \leq t_{opt} \leq \frac{2n \|\theta^*\|_2^2}{p\sigma^2 + (p - d) \|\theta^*\|_2^2}.$$

Theorem 2 approximates the early optimal stopping $t_{opt}$ up to a constant when $p$ is larger that $n$ and $d$ by a constant factor. For $d$ that is close to $p$, numerical experiments show that the approximation

$\tilde{\Theta}\left(\frac{n\|\theta^*\|_2^2}{p\sigma^2+(p-d)\|\theta^*\|_2^2}\right)$ still holds. Theorem 2 implies that **when the number of features is larger than the model dimension, optimal early stopping time increases as $n$ or $d$ increases.** An intuitive explanation is when the model is when the model size and the sample size are small, the model does not have enough information to train so that the model tends to fit the noise easier.

**Proposition 2.** *In setting $\mathcal{S}_{under}$, for a fixed parameter $\theta^*$ and noise variance $\sigma^2$, let $\bar{t}_{\mathrm{opt}} = \frac{n\|\theta^*\|_2^2}{p\sigma^2+(p-d)\|\theta^*\|_2^2}$. Then, when $(8n+9d+16)\|\theta^*\|_2^2 \le p\sigma^2 + p\|\theta^*\|_2^2$, we have*

$$\overline{R}_1 \le \overline{R}_{\theta^*}(\bar{t}_{\mathrm{opt}}) \le \overline{R}_2,$$

*where $\overline{R}_1 = \sigma^2 + \|\theta^*\|_2^2 - \frac{2nd\|\theta^*\|_2^4}{p^2\sigma^2+p(p-d)\|\theta^*\|_2^2}$ and $\overline{R}_2 = \sigma^2 + \|\theta^*\|_2^2 - \frac{3nd\|\theta^*\|_2^4}{4\left(p^2\sigma^2+p(p-d)\|\theta^*\|_2^2\right)}$.*

*Here, $\overline{R}_1 \ge 0.8\overline{R}_2$ and both $\overline{R}_1$ and $\overline{R}_2$ decrease when $d$ and $n$ increase.*

Proposition 2 gives an upper bound and a lower bound on the risk at the approximated optimal early stopping time. Under the assumption $d$ and $n$ are small, $\overline{R}_1$ and $\overline{R}_2$ give a tight region ($\overline{R}_1 \ge 0.8\overline{R}_2$). Proposition 2 supports the observation that **risk at optimal early stopping time monotonically decreases as $n$ and $d$ increases**. Numerical experiments are presented in Section F.1.

## 4  EXPERIMENT

In this section, we run simulations for settings $\mathcal{S}_{\mathrm{over}}$ and $\mathcal{S}_{\mathrm{under}}$ and demonstrate that training deep neural networks also follows the trends our theoretical analysis identifies. We provide additional experimental results in Appendix F.

### 4.1  ~~LINEAR REGRESSION~~

 (We moved the experiments results on linear regression to the appendix so we can have more space to address the comments from the reviewers.)

### 4.2  DEEP LEARNING WITH NEURAL NETWORKS

We study the optimal early stopping time in neural network training. We demonstrate that datasets MNIST, CIFAR-10 and CIFAR-100 have low-rank representations. Such representations show that training common deep neural networks on these datasets falls under the overparametrization setting. Then, we demonstrate experimentally that the optimal early stopping time follows the pattern we identify for the overparametrization setting.

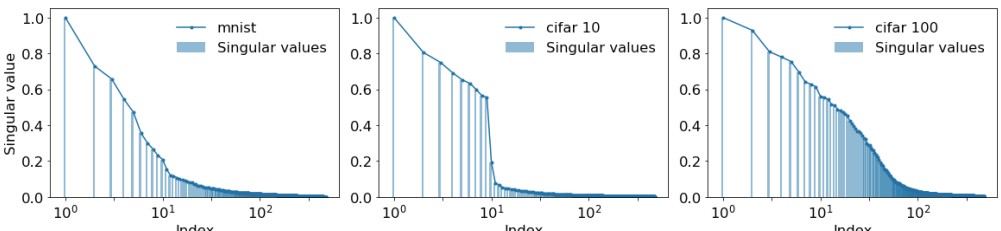

Figure 3: Scaled singular values of representations obtained from MNIST, CIFAR-10 and CIFAR-100.

**Existence of low-rank structure in image classification.**    Many previous works observe that low-rank structures exist and can be leveraged in deep image classification tasks (Van der Maaten & Hinton, 2008; Dusenberry et al., 2020). This structure facilitates the self-supervised learning and transfer learning paradigm where image features are learned and fixed close to the input layer, reducing the complexity of image-to-label mapping. Then only the last layer is trained to adapt to and classify out-of-domain images (Chen et al., 2020b). We follow this approach and demonstrate the existence of the low-rank structures in Figure 3, where we plot the singular values of the representations of MNIST, CIFAR-10, and CIFAR-100. First, we train CNN/ResNet-60 (He et al., 2016a;b) to convergence with zero training loss (no label noise). Then, we obtain the representations produced by the trained neural networks on MNIST, CIFAR-10, and CIFAR-100. Even when trained with a very wide neural network, only the first 10/100 singular values of the representations are significant

compared to the others. This suggests that low-rank feature representations of the datasets exist. We believe this phenomenon may be of independent interest to understanding neural networks.

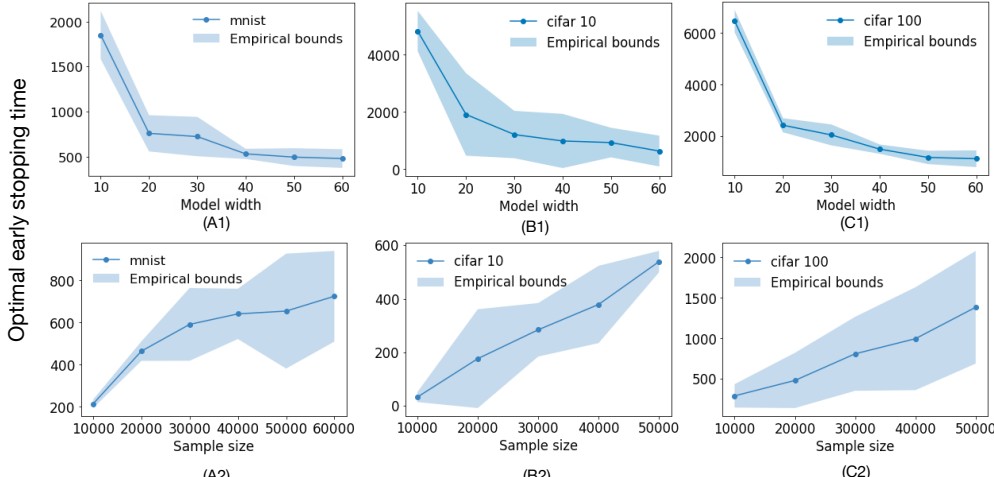

Figure 4: In (A1) (B1) (C1), optimal early stopping time decreases with increasing model sizes for MNIST, CIFAR-10, and CIFAR-100. In (A2) (B2) (C2), optimal early stopping time increases with increasing sample sizes for MNIST, CIFAR-10, and CIFAR-100.

**Overparameterization behavior in image classification using neural networks**    In this section, we study the optimal early stopping time of MNIST, CIFAR-10, and CIFAR-100. We train MNIST using a fully connected network, and CIFAR-10/100 using ResNet. Part of our code in this part builds on open source code from (Nakkiran et al., 2020a). We add $20\%$ of label noise to all datasets For MNIST, a simple ReLU network with one hidden layer is used. We vary the hidden layer width over $d = [10, 20, ..., 60]$. For CIFAR-10/100, we use a family of ResNet networks with convolutional layers of widths $[d, 2d, 4d, 8d]$ for different layer depths. The sample size ranges from $n = [1e^5, 2e^5, ..., 6e^5]$ for MNIST, and $n = [1e^5, 2e^5, ..., 5e^5]$ for CIFAR-10/100. Inputs are normalized to $[-1, 1]$ with standard data-augmentation. We use stochastic gradient descent with cross-entropy loss, learning rate $\eta = 0.1$ for MNIST, CIFAR-10, and $\eta = 0.05$ for CIFAR-100, and minibatch size $B = 512$. The optimal stopping time is computed as $t_{opt} = \eta T \frac{n}{B}$, where $T$ is the epoch with the optimal early stopping performance. To select the optimal stopping time, we first apply a moving averaging window with length 5 to mitigate the effect of mini-batch noise. After smoothing, we select the point with the optimal stopping time with the first minimal testing error (the lowest point of the first 'valley', ignoring small local minima).

We show in Figure 4 the optimal early stopping time for all three datasets with varying $n$ and $d$. In (A1) (A2) (A3), we observe that the optimal early stopping time decreases with increasing model width. In (B1) (B2) (B3), we observe that the optimal early stopping time increases with larger sample sizes. Due to the existence of low-rank representations, Figure 4 falls under the overparametrization setting even for small $d$. The observations are consistent with our theoretical analysis and the experimental results on linear models in the overparametrization setting. When trained with ultra small $d < 10$, the models show great instability. We provide additional experimental results supporting the underparametrization setting in Section 4.3.

## 4.3    CLASSIFICATION USING TWO-LAYER NEURAL NETWORKS

Training datasets like CIFAR-10/100 using ultra small models is unstable, so in this section, we use simpler datasets to verify our theoretical results for the underparametrization setting. In this experiment, we use *make_classification* function in *sklearn* (Pedregosa et al., 2011) to generate clusters of points normally distributed around vertices of a high-dimensional hypercube and assigns an equal number of clusters to each class. We generate 2000 samples (50-50 train-test split) of 20 dimensions/features, but only $p = 10$ dimensions/features of each sample are informative features. We add $20\%$ label noise to the dataset. The optimal stopping time is selected similarly as in Section 4.2.

We train the data using two-layer ReLU networks with varying hidden layer width $d$. The blue area in Figure 5 gives $0.5$ standard deviation of 6 runs and the blue line gives the mean. In Figure 5a, when $d \leq p = 10$, we observe the underparamatrization behavior where the optimal early stopping time increases as the model size $d$ increases. When $d \geq 10 = p$, the optimal stopping time decreases with $d$. We can further observe that the test loss decreases with increasing model sizes at the optimal early stopping time.

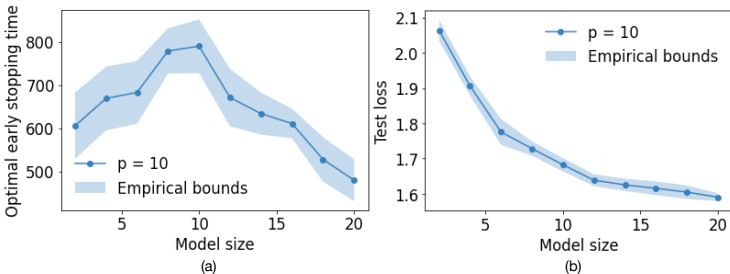

Figure 5: (a) Optimal stopping time as a function of model size. (b) Test loss as a function of model size.

## 5 CONCLUSION AND FUTURE WORK

In this paper, we propose a new model of overparametrization to study the optimal early stopping time. This model captures the phenomenon that the model size usually exceeds the number of features in practice. We believe this model of overparametrization can be of independent interest. One future direction is the performance of other optimizers or techniques, e.g., stochastic gradient descent or data augmentation, on this model. Moreover, we give an explicit characterization of the early stopping time and the resulting model. Future works can explore other regularization properties of this early stopped model, e.g., whether this early stopping helps when there is a domain shift in the testing dataset.

Another direction worth exploring is to extend our results to nonlinear settings. Extending our result to the lazy training regime of neural networks (e.g., random feature regime and neural tangent kernel regime) can be relatively straightforward. One approach is to first locate the features extracted at initizalition and then show how gradient flow progresses on the features extracted. Theoretical analysis of the non-lazy training is still a largely open question, so extending this result to non-lazy setting can be more challenging. To extend this result to the non-lazy setting for a fully nonlinear neural network model, one needs to study how the features extracted change and how the weights on the features change at the same time.

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
