# OpenReview forum: "On Optimal Early Stopping: Overparametrization versus Underparametrization"
_ICLR.cc/2022/Conference — ICLR 2022 Submitted_

### Official Review · Reviewer_Y1pf · 2021-10-31

**Correctness:** 4
**Technical Novelty And Significance:** 2
**Empirical Novelty And Significance:** 3
**Recommendation:** 5
**Confidence:** 4

**Main Review:**

I think the main contribution of the paper is the analysis of optimal early stopping in overparameterized regime. However, I think the definition of “overparameterized regime“ in this paper is different from what is usually used in the literature. Specifically, the definition in the paper doesn’t take into account the number of examples, but only the number of features and model size. I think the authors should clarify what do they mean by “model size” in general (e.g. in neural networks), is it the number of parameters? The width of the network ?

Another relevant paper in the overparameterized regime:

Vaskevicius et al, Implicit regularization for optimal sparse recovery. NeurIPS 2019.

However, I think the novelty of the paper is quite limited. The underparametererized setting was previously analyzed in Nakkiran et al., 2020b, where the effect of regularization is studied, and it is well known that $L_2$ regularization is strongly related to early stopping. For example, Nakkiran et al., 2020b showed that the expected test risk of the optimally-regularized model is monotonic in the model size (theorem 3 therein). Also, the optimal ridge parameter $\lambda\sim\frac{1}{d}$ (lemma 3 therein), and since stopping time decreases when $\lambda$ increases, the optimal stopping time is monotonic in model size, which is also the result in this paper. The authors claim that “optimal early stopping can help mitigate double descent in various settings” – this should be compared to similar results in Nakkiran et al., 2020b for $L_2$ regularization.

Comments and questions about the empirical part:

I think the empirical part with neural networks should be improved:

-	Is it possible to show that the results hold when changing the model size by changing the depth and not width ?
-	Would be good to show that the results hold when playing with the amount of label noise.
-	I think Figure 13 should go more to the right (more epochs) to see the entire curve.
-	In figures 11,12 – can you explain how $a,b$ are selected ?
-	In figure 14 – is the test loss correspond to optimal early stopping time ?


**Summary Of The Paper:**

In this paper the authors analyze early stopping in linear regression. The authors show that the optimal early stopping time behaves differently in overparameterized and underparameterized regimes. Specifically, the optimal early stopping time decreases when increasing model size in the overparameterized regime, but increases with model size in the underparameterized regime. Some empirical results demonstrate the theoretical results in linear regression and deep neural networks.

**Summary Of The Review:**

Overall, I think the theoretical contribution is rather limited given the work of Nakkiran et al., 2020b, and the empirical part should be improved. Therefore, I don’t recommend the paper for acceptance at this stage.

---

> ### Author Response · Authors · 2021-11-17
> **Response to Reviewer Y1pf**
>
> Thank you for your time spent and your feedback.
>
> 1. “The underparametererized setting was previously analyzed in Nakkiran et al., 2020b, …. Also, the optimal ridge parameter , and ...the optimal stopping time is monotonic in model size, which is also the result in this paper.”
>
> We first want to clarify that the main result of our paper is to show that the early stopping time behaves very differently in the setting where the number of features exceeds the model size and in the opposite setting. The results on risk monotonicity are only byproducts of the main result. Also, the overparameterized setting studied, which better reflects what happens in practice, and studying this problem in these two regimes are new and previously underexplored.
>
> Even for the underparametrized setting, what we show differs significantly from previous works. In this underparametrized setting, stopping at $t$ is equivalent to using a regularization term in the form of $\beta^\top Q_t\beta$ for a matrix $Q_t$ that depends on the data. Nakkiran et al., 2020b studied the simpler L_2 regularization in the form of $\lambda ||\beta||^2$. Nonlinear nature of this parametrization renders the techniques developed in Nakkiran et al., 2020b not applicable in our early stopping setting and we don’t think results in Nakkiran et al., 2020b can be straightforwardly transferred to our case.
>
> ** We would appreciate it if the reviewer can reevaluate the novelty of our paper given the above clarification. **
>
> 2. “I think the authors should clarify what do they mean by “model size” in general (e.g. in neural networks), is it the number of parameters? The width of the network ?”
>
> In our paper, “model size” means how many features the model is able to extract. For example, in the lazy training regime, where the random initialization decides the feature, wider neural networks will be able to see more features. Then, our “model size” is the width of the neural network. For general neural networks, we think the width of the networks is more important than the depth in deciding the number of features extracted. The depth relates more to the complexity of the features extracted. However, understanding fully nonlinear neural networks is still an open problem, so we are open to the different answers to the question. We will add the above discussion to the revised paper. Thank you for asking this question and we hope the above discussion is helpful.
>
> 3. Thank you for the relevant paper you pointed out. We have cited the paper in our revised version.
>
> 4. We added the following additional experimental results in Appendix F.2 in the revised paper. Due to time constraints, we only run each experiment once.We will keep refining those results.
>
> - “Is it possible to show that the results hold when changing the model size by changing the depth and not width ?” - We have added some experimental results on this in Appendix F.2. The experiments show that the optimal early stopping time decreases with increasing network depth.
>
> - “Would be good to show that the results hold when playing with the amount of label noise.” - We have added some experimental results on this in Appendix F.2. The experiments show that our observations are consistent when changing the amount of label noise.
>
> - “I think Figure 13 should go more to the right (more epochs) to see the entire curve.” - We will run the experiments in fig 13 longer in the revised version.
>
> - “In figures 11,12 – can you explain how  a,b are selected?” - Since our theoretical result only shows the bound up to $\Theta$, a,b here are selected as the reasonable constants so that we can show the empirical trends are consistent with the theoretical trends.
>
>  - “In figure 14 – is the test loss correspond to optimal early stopping time ?” - Yes
>
> Thank you for you suggestions on the experiment parts. We hope you find our additional experimental results helpful and we will add more results in our final version.

---

> > ### Comment · Reviewer_Y1pf · 2021-11-26
> > **Response**
> >
> > Thank you for the feedback.
> >
> > I also read the other reviews (and answers) and decided to keep my score as is.
> >
> > I think the paper requires at least one additional revision before publication. My main issues are:
> >
> > 1. The definition of over/under parametrization and model size. I agree with reviewer EU5H that in fact, a more appropriate naming is "over/under informative". In addition, the definition of model size as "how many features the model is able to extract" is not so clear and well defined. Also, it is not so clear what "inherent features of the data" are for some dataset.
> >
> > 2. Detailed comparison to Nakkiran et al., 2020b, explaining the differences and why the results of Nakkiran et al., 2020b cannot be transferred to the setting in this paper.

---

> > > ### Author Response · Authors · 2021-11-27
> > > **Response to Reviewer Y1pf**
> > >
> > > Thank you for your response.
> > > - We will change the wording "over/under-parametrization" in our revised paper. Thank you for your and Reivewer EU5H's suggestion.
> > > - There are a huge number of different neural networks used in practice, so we don't think it is possible to have a unified model size definition for all of them. As we wrote in the previous response, understanding fully nonlinear neural networks is still an open problem. For simple neural networks, e.g., shallow fully connected neural networks, we believe "model size" is the model width. We agree that it is not possible to show the inherent features of every dataset, but we showed experimentally the feature number of some of the most popular vision datasets in Section 4.2.
> > > - We will add the comparison to Nakkiran et al., 2020b to the revised paper, but as we have pointed out, what we show differs significantly from Nakkiran et al., 2020b even in the underparametrization(underinformative) setting.

---

### Official Review · Reviewer_6rDq · 2021-11-01

**Correctness:** 3
**Technical Novelty And Significance:** 3
**Empirical Novelty And Significance:** 3
**Recommendation:** 6
**Confidence:** 3

**Main Review:**

*Strong point(s) of the paper*

- The authors obtain novel theoretical results on the optimal stopping times for gradient flow under a linear model (with Gaussian data/noise assumptions), refining the initial theory built up by Ali et al. (2019).

- Their theoretical analysis explicitly separates the over-parametrized and under-parametrized settings (e.g., whether the number of model parameters in larger than the number of data features or not), obtaining distinct insights for each of these settings, with empirical tests that are designed to invesigate the "sharpness" of this distinction. This distinction is, at least in extreme cases, quite important, and additional insights both in theory and practice are of value.


*Weak point(s) of the paper*

- I felt that the theoretical results were poorly explained, given the context of previous work such as Ali et al. (2019). In section 2.1, the authors just say "we can derive a high probability upper and lower bound on (the optimal stopping time)" and give the result, without any discussion of the technical side of this. Since Ali et al. say that controlling the optimal stopping time is "difficult" (their Rmk 4), one would expect that the authors here would describe what assumptions or techniques made it possible for them to solve this difficulty. Since this previous work did not (as far as I can tell) make any Gaussian assumptions on the data, one expects that the Gaussianity might play an important role in the authors' analysis, but since this is not described, the reader is left wondering. One is also left wondering if it would be straightforward to obtain high-probability bounds on the risk, since bounds are with high probability for the stopping time control.

- While I would not say the writing itself is poor, it is rather careless in many places. Some are notation issues, others are explanation issues. Here are a few representative examples:
  - The matrix exponential (p.4) is not introduced anywhere as far as I can see.
  - On p.6, the authors say that "Theorem 2 implies that when the number of features is larger than the model dimension, optimal early stopping time increases as $n$ and $d$ increases." This is a very broad statement; in reality, doesn't it depend on how fast $n$ and $d$ respectively increase? A similar statement is on p.7 for the under-parametrized case.
  - In the experimental section, there is very little information on how the optimal stopping times are actually computed, and a brief glance at the supplementary materials does not shed any more light on this.


**Summary Of The Paper:**

This work looks at implicit (algorithmic) regularization of learning algorithms, with a particular focus on the impact of early stopping in the context of iterative gradient-based learning algorithms.

The model of interest is a linear model with additive random noise, and for their analytical entry point, the authors consider the gradient flow differential equation as in Ali et al. (2019), i.e., a typical gradient descent update in the limit where step size is taken to zero. They provide confidence intervals for the optimal early stopping time (optimal in terms of the expected squared error) and subsequently derive upper/lower bounds in expectation for the risk achieved at such the optimal early stopping time.

In addition, the authors carry out empirical analysis of optimal stopping times for both linear and non-linear models, over a range of sample size, data dimension, and model dimension settings. They argue that well-trained neural networks admit "low-rank" representations which are akin to the over-parametrized linear model that they study in theory.

**Summary Of The Review:**

To the best of my understanding, the theoretical results obtained by the authors (optimal stopping time control + resulting risk bounds) are new and provide more refined insights that the existing literature with respect to the impact of sample size and dimensionality, despite the limiting Gaussian assumptions. In addition, while the experimental section could use a lot more exposition, from what I can tell the design is straightforward and natural considering the theoretical results the authors want to verify. My overall take on this paper is that there are some results of value here, but the paper itself needs some work, so I am left on the borderline with this work, tending to accept, but with the caveat that I am not very familiar with the developments in this line of work since Ali et al.

---

> ### Author Response · Authors · 2021-11-17
> **Response to Reviewer 6rDq**
>
> Thank you for your time spent and your feedback.
>
> 1. Part of the results in Ali et al. (2019) assumed that the data follows gaussian distribution. There was no explicit formula for the optimal stopping time because the eigenvalues of the data matrix were not explicitly characterized.
>
> 2. “On p.6, the authors say that ...This is a very broad statement” What we meant here is increasing one of n and d and keeping other parameters fixed, the optimal early stopping time increases.
>
> 3. “The matrix exponential (p.4) is not introduced.” We have added the definition of exp to the revised version.
>
> 4. “​​how the optimal stopping times are actually computed.”
> For linear regression data with gradient flow, we select the optimal stopping time with the first local minimal loss. This is easy to observe since the error vs. epoch curve is smooth (like Figure 6). For classification settings with SGD, we first apply a moving averaging window with length = 5 to mitigate the effect of mini-batch noise. After smoothing, we observe test error vs. epoch curve like Figure 11. We then select the point with the optimal stopping time with the first minimal testing error (the lowest point of the first ‘valley’, ignoring small local minima). We feel that this post hoc selection is time consuming and hope that our theoretical analysis can be helpful for practitioners. We have added this clarification to the revised version.
>
> Thank you for your valuable suggestions on writing. We will keep revising the paper to make sure that notations are defined and statements are clear.

---

### Official Review · Reviewer_EU5H · 2021-11-02

**Correctness:** 3
**Technical Novelty And Significance:** 2
**Empirical Novelty And Significance:** 2
**Recommendation:** 3
**Confidence:** 4

**Main Review:**

# Pros:
+ The different behaviors of optimal stopping time for gradient flow in over- or under-informative settings seem to be interesting and novel.
+ The observation from neural network experiments are also inspiring.


# Cons:
- First of all, the terminology of "overparameterization" and "underparameterization" are rather mis-leading!!!! In literature, those terminologies are widely used to refer to regimes where the number of parameter is larger/smaller than the number of data. However in this paper, if I understand it correctly, the authors aim to mean two different feature generating processes. More concretely, let $X$ be feature, $Z$ be the hidden variable and $Y$ be the response, then the two regimes in Markov chain can be represented as
$$
X \\to Z \\to Y
$$
and
$$
X \\leftarrow Z \\rightarrow Y
$$
I would rather refer these two regimes as "over-informative" and "under-informative". Better to avoid misleading here.

- Beyond terminologies, I would say the paper needs some further revision, at least in terms of languages. For a few examples:
   * the first two paragraphs in Sec 1 could use some references.
   * Sec. 1, Over- vs. Under-parameterization paragraph: I am not sure whether or not these are "discovered" by the authors. There are a number of references in Sec. 4.2 that seem to suggest low-dim data structure is known. Could use some clarification here.
   * The paragraph above Sec. 1.1. "early-stopped models"
   * It is such a mess when introducing $P$ --- is it deterministic or random?

- Paragraph below Thm1. The claims about case $ n \gg d$ is clearly incorrect. Note that $n / (d+n)  = \Theta(1)$, but $\log (1 + \Theta(n / d)) \gg 1$, so the logarithmic factors are actually leading factors. All the following claims need revision.

- As for the Theorems, I find the feature assumption is too strong to be interesting. For example in Thm1, it assumes $x \sim N(0,1)$. This is too specialized and is basically equivalent to a 1-dimensional problem. Thm1 could be more interesting if can cover $x \sim N(0,H)$ where $H$ is a general PSD matrix. Similar assumptions on hidden variable is also made for Thm2.

- Finally, I feel the main conclusion of this work is less surprising (or interesting), given the fact that the different behaviors of optimal early stopping is in fact caused by **different data generating process** instead of **different regimes of number of parameter vs. number of data**. In the first setting of the paper, which is also the traditional setting we see in literature, the feature contains full information for recovering the label, and the optimal stopping time is obtained by balancing the bias vs. variance error. In this case, it is very clearly one needs to stop earlier with a smaller $n$ or larger $d$ to avoid overfitting. In the second setting, it is more like an issue of insufficient observation, and no wonder one can stop latter as $d$ increases (where the information in features increases). I do not think this is beyond the traditional statistics wisdom.









**Summary Of The Paper:**

This paper studies the optimal early stopping time of gradient flow for linear regression in two regimes: over-informative features or under-informative features, where the first regime means the features contain sufficient information to generate the response, while the second regime means responses rely on additional information beyond those in the features. In the two different regimes, the paper shows that the optimal early stopping time behaves quite differently. Theory are built for linear regression, and experimental verifications are performed in both toy data and neural networks.

**Summary Of The Review:**

The current paper is misleading and needs a significant revision. Based on my current understanding, the paper is less interesting as suggested by the title. I cannot recommend acceptance given the current version.

---

> ### Author Response · Authors · 2021-11-17
> **Response to Reviewer EU5H**
>
> Thank you for your time spent and your feedback.
>
> 1. “the different behaviors of optimal early stopping is in fact caused by different data generating process instead of different regimes of number of parameter vs. number of data…I do not think this is beyond the traditional statistics wisdom.”
>
> We first want to clarify that in the linear model we study in the theory part, the input $X$ is not the traditional input data people studied in the past. The best way to think about this linear model is to think of it as the last linear layer of neural networks. The input $X$ is the features the neural networks extract and train on. Then, a wide neural network tends to extract more features than a narrow neural network, even when the inputs to the two neutral networks are the same. The two regimes we study are the settings where the model extracts more features than the number of inherent features of the data and the opposite setting.
>
> The data are generated differently in the two different regimes because we want to capture the difference between the two settings we study. Note that when the number of features equals the model size, those two regimes agree with each other. In our paper, we focus on studying the number of features instead of the number of samples because we believe whether the model size exceeds the number of features is a better criterion in determining how the optimal early stopping time behaves. The experimental results also corroborate this finding. We believe studying this problem from the viewpoint of feature number is one of our main contributions.
>
> ** We hope the above discussion clarifies the reviewer’s misunderstanding of our work and would appreciate it if the reviewer can reevaluate our contributions. **
>
> 2. “the terminology of ‘overparameterization’ and ‘underparameterization’ are rather mis-leading”.
>
> We think the word “overparameterization” means given the model and the data, there are more than one set of parameter values that can fit the training data perfectly so it is worth studying which set of parameter values the training algorithm leads to. We don’t think “overparameterization” words can only mean “the number of parameters is larger/smaller than the number of data” in the literature. However, we understand that those two words are more often used when comparing number of parameters and number of data points, so we are willing to change these two words if the reviewers agree that the current wordings are misleading.
>
> 3. “the first two paragraphs in Sec 1 could use some references.”
>
> We have added some references to Sec 1.
>
> 4. “Sec. 1, Over- vs. Under-parameterization paragraph”
>
> We are not aware of any work that studies the exact same setting as we do. We changed the word to “noticed” to avoid any confusion.
>
> 5. “Is P deterministic or random?”
>
> P is random. We stated “We further assume that P is a uniformly random semi-orthogonal matrix” when introducing the two settings in our paper.
>
> 6. “Paragraph below Thm1....”
>
> The paragraph below thm1 is omitting the logarithmic dependence. However, we agree that in the case when $n>>d$, the log term matters. This won’t affect the following claims that when the number of features is smaller than the model dimension, optimal early stopping time increases as n increases and d decreases. We added this discussion to the revised version. Thank you for pointing this out.
>
> 7. “I find the feature assumption too strong to be interesting. For example in Thm1, it assumes x∼N(0,1). This is too specialized and is basically equivalent to a 1-dimensional problem.”
>
> We don’t understand why this setting is equivalent to a 1-dimensional problem. We assume $x\sim \mathcal{N}(0,I_d)$. Our results heavily rely on that our model and data are high-dimensional. We will really appreciate it if the reviewer can clarify what they mean here.
>
> 8. “Thm1 could be more interesting if can cover x∼N(0,H). ”
>
> Our result can be extended to $x∼N(0,H)$ straightforwardly as long as we can bound the eigenvalues of the data matrix $X$ (see Lemma 2 and Lemma 3).

---

> > ### Comment · Reviewer_EU5H · 2021-11-27
> > **Thanks for the reply**
> >
> > Thanks for the reply.
> >
> > * [Over/under-parameterization] I agree one can have different understanding on the word "over/under-parameterization". However, as a scientific terminology in ML theory papers, it has already been widely adopted to refer to the regime that "number of data $<$ or $>$ number of parameter", e.g., see [1,2] and their follow-up works. Just in order to differentiate the setting of this work to the existing ones, as well as to avoid unnecessary reloading of definitions, a better idea in my perspective is to avoid using the word "over/under-parameterization" in this paper.
> >
> > * [Is P deterministic or random] To elaborate my initial comment on this, please allow me to cite several sentences from the original paper:
> >    - "For any fixed dataset $y \\in R\^n$ and $X \\in R\^{n\\times d}$ given by **fixed P** ..."
> >    - "We further assume that $P\\in R^{p\\times d} $ is a uniformaly **random** semi-orthogonal matrix".
> >
> >    These sentences contradicting to each other in my perspective thus I do not understand whether or not $P$ is random or not. I guess the authors could describe the setting with conditional probability, which is more clear and rigorous.
> >
> > * [logarithmic factor] Thanks for confirming this typo.
> >
> >
> > * [$X \sim N(0, I)$] Sorry for being sloppy in my initial comment here. I agree with the authors that dimensionality factor is important as there are concentrations inequalities applied. My biggest concern is that the assumption $X \\sim N(0, I)$ implies an **isotropic** input distribution where every direction will be symmetric in the analysis. For example, Lemma 2 & 3 suggest that the empirical covariance/gram matrix is nearly an identity after a properly scaling. However, if one allows $X \\sim N(0, H)$ where the Gaussian is **anisotropic**, the concentration is much more subtle as one might not get sufficient data to bound the spectrum of the empirical covariance/gram matrix around the small eigenvalue directions (see [1] for some discussions). If the authors could elaborate more on how to extend the current theory to $X \\sim N(0, H)$, even under some additional (but none trivial) condition on $H$, the results would be much more interesting in my perspective.
> >
> > * In additional, I think polishing the general writing of this paper could be helpful.
> >
> >
> >
> > [1] Bartlett, Peter L., Philip M. Long, Gábor Lugosi, and Alexander Tsigler. "Benign overfitting in linear regression." Proceedings of the National Academy of Sciences 117, no. 48 (2020): 30063-30070.
> >
> > [2] Belkin, Mikhail, Daniel Hsu, and Ji Xu. "Two models of double descent for weak features." SIAM Journal on Mathematics of Data Science 2, no. 4 (2020): 1167-1180.

---

> > > ### Author Response · Authors · 2021-11-29
> > > **Response to Reviewer EU5H**
> > >
> > >  Thank you for your suggestions.
> > > - We will change the wording "over/under-parameterization" in the revised paper.
> > > - In the sentence "For any fixed dataset given by a fixed P, let $\beta_{X,y}(t)$ be ...", what we mean is $\beta_{X,y}(t)$ is defined on a fixed realization of X, y and P.
> > > - In our paper, to find the optimal early stopping time, we find $t^*$ such that $dR/dt =0$. Then, we study how $t^*$ scales with $d$ (and $n$) by studying how the eigenvalues of the matrix $\frac 1 n X^\top X$ scales with $d$ (e.g., see the proof of Theorem 1). The key part is  how the eigenvalue scales as we change d instead of how the eigenvalues concentrate around its expectation. In the case when $n<d$, it is possible that the eigenvalues of the data matrix $\frac 1 nX^\top X$ don't have good concentration, but we should still expect that the non zero eigenvalues increase when we increase $d$. For example, consider the matrices $X\in \mathbb{R}^{n\times d}$ and $X'\in \mathbb{R}^{n\times (d+1)}$ which differ by only one column, both of them have $n$ non-zero eigenvalues but $tr(\frac 1 nX'^\top X')\geq tr(\frac 1 n X^\top X)$.

---

### Official Review · Reviewer_dczT · 2021-11-02

**Correctness:** 4
**Technical Novelty And Significance:** 3
**Empirical Novelty And Significance:** Not applicable
**Recommendation:** 6
**Confidence:** 3

**Main Review:**

I quickly checked the proof and did not find an obvious flaw.

Pros:

1. The problem is interesting and important. Early stopping has a long history of being used as a regularization of machine learning algorithms.

2. The results are insightful for me. The main results about the overparameterization setting agree with some empirical findings and make a step towards understanding the training procedure of machine learning algorithms.

Cons:

1. The data distribution and problem are restrictive. I understand that this is an early theoretical attempt and I notice that the authors also find some related work falls into the same setting. However, the approaches in this paper rely on the assumption that the estimation can be expressed as a function of early stopping time. I worry that it would be highly nontrivial to extend the results to real settings.

2. Technically, the results of Theorem 1 and 2 are not directly comparable because Theorem 1 presents high probability bounds while Theorem 2 presents deterministic bounds with a further assumption on the data distribution. It may be confusing to present the main claims in parallel without discussing the differences in the two settings.

**Summary Of The Paper:**

This paper studies an interesting and important problem: what is the optimal early stopping time (considering the model dimension and sample size, especially under an overparameterization setting)? The data distribution is synthetic. Considering a standard linear least squares regression problem, the gradient flow is expressed as a differential equation. Further, solving the differential equation, the estimator can be expressed as a function of the time and data. Given such a result from an earlier work, the paper studies the optimal stopping time that minimizes the expected risk and presents high probability (upper and lower) bounds for the overparameterization setting. It suggests that with a high probability, the optimal early stopping time is $\tilde{\Theta}(\frac{n}{n+d})$, which implies that the optimal early stopping time increases as $n$ increases and $d$ decreases. The paper also shows that the bounds decrease as $n$ increases, which implies that optimal early stopping can mitigate sample-wise double descent. Empirical results validate the theory.

**Summary Of The Review:**

Overall this is an interesting theoretical paper with its limitation. Currently, I tend to accept it.

---

> ### Author Response · Authors · 2021-11-17
> **Response to Reviewer dczT**
>
> Thank you for your time spent and your feedback.  We really appreciate that you find our results insightful.
>
> 1. “The data distribution and problem are restrictive.”
>
> The main goal of our paper is to present our observation that the early stopping time behaves differently when the number of features exceeds the model size and vice versa, and explain this phenomenon theoretically. We studied the simplest model that can explain such behavior and convey our main idea. It is possible to extend our results to the non-linear models in the lazy training regime (e.g., random feature regime and neural tangent kernel regime). We can first locate the features extracted at initizalition and then show how gradient flow progresses on the features extracted, similar to what our paper currently does. Extending our paper to such settings will make the proof more complicated, but we don’t think it will provide much more insight than what our paper has already shown, so we decided to stick to our current model. However, we agree that studying general non-linear models is a direction future works can explore.
>
> 2. “Technically, the results of Theorem 1 and 2 are not directly comparable.”
>
> Those two results are in slightly different forms because we use different techniques to prove the two theorems. We found those two forms to be the most natural results obtained through the two techniques. A high probability bound like what we show in Theorem 1 is in general stronger than an expectation bound, so we believe it is fair to compare Theorem 1 and 2.

---

### Decision · Program_Chairs · 2022-01-20

**Decision:**

Reject

**Comment:**

This paper studies the problem of characterizing the optimal early stopping time in overparameterized learning as a function of model dimension and sample size. To do this the paper uses an explicit form of the gradient flow from prior work to present high probability bounds in the over-parameterized setting and characterizes various properties of the optimal stopping time. The authors also conduct various experiments to verify the theory. The reviews though the paper was interesting and insightful. They also raised some concerns about the (1)restrictiveness of the distributional assumptions, (2) poor explanation of the theoretical results, and (3) novelty with respect to other work and (4) other technical issues. The discussion and response mitigated these concerns but the reviewers decided to mostly keep their original score. My own reading of the paper is that there are good ideas in this paper and I agree with the authors that some of the technical issues raised by the reviewers is incorrect. However, it is also clear that the paper needs a bit more work to put it into the right context and also the proof need to be more clearly and carefully written before this paper can be accepted. Therefore I recommend rejection but encourage the authors to submit to a future ML venue after a thorough revision.